Prevalence of anxiety, depression, stress and insomnia among healthcare professionals during COVID-19 in Africa: umbrella review of existing meta-analyses

Hasen Aragaw Asfaw aragawasfaw5@gmail.com 1
Mohammed Ahmed Adem 2
Seid Abubeker Alebachew 2
1 Department of Statistics, College of Natural and Computational Sciences, Samara University , Semera , Afar , Ethiopia
2 Department of Nursing, College of Medicine and Health Sciences, Samara University , Semera , Afar , Ethiopia
Basu Arindam
Electronic publication date: 2024 Oct 30
Publication date: 2024
Volume: 12
Electronic Location ID: e18108
Received 2024 Feb 12; Accepted 2024 Aug 27
Copyright: ©2024 Hasen et al.
Copyright year: 2024
Copyright holder: Hasen et al.
License: This is an open access article distributed under the terms of the Creative Commons Attribution License, which permits unrestricted use, distribution, reproduction and adaptation in any medium and for any purpose provided that it is properly attributed. For attribution, the original author(s), title, publication source (PeerJ) and either DOI or URL of the article must be cited.
License URL: https://creativecommons.org/licenses/by/4.0/

Keywords: COVID-19, Mental health, Mental disorders, Healthcare professionals, Anxiety, Depression, Stress, Insomnia, Umbrella review, Africa

Funding: The authors received no funding for this work.

==============================
Introduction

In Africa, healthcare professionals experienced various mental health problems during COVID-19. However, very little was done on the extensive evidence regarding mental disorders. The purpose of this umbrella review is to provide comprehensive data on the prevalence of anxiety, depression, stress, and insomnia among healthcare professionals during the COVID-19 pandemic in Africa.

Materials and Methods

Systematic searches of databases African Journals Online, MedRxiv, PubMed, and Google Scholar were used to identify studies from the occurrence of COVID-19 from December 2019 to March 2023 were included. To pool the gathered data for results with a 95% confidence interval (CI), DerSimonian-Laird random effects meta-analysis was used. For heterogeneity examination, I2 was used. The quality assessment was evaluated by using the Joanna Briggs Institute (JBI) critical appraisal checklist.

Results

A total of five studies reported the prevalence of depression, the pooled prevalence was 53.75% (95% CI [40.80–66.70], I2 = 63.6%, p = 0.027). In a total of four studies, the pooled prevalence of anxiety was 49.97% (95% CI [34.71–65.23], I2 = 71.26%, p = 0.014). From a total of two studies, the pooled prevalence of stress was 57.27% (95% CI [42.28–72.25], I2 = 58.9%, p = 0.119). From a total of four studies, the pooled prevalence of insomnia was 45.16% (95% CI [32.94–57.39], I2 = 50.8%, p = 0.107).

Conclusions

The COVID-19 pandemic highly affects the mental health of healthcare professionals in Africa. Stress, depression, anxiety, and insomnia symptoms were representing the most common based on evidences from existing meta-analyses. This evidence can help experts when executing specific interventions that address mental health problems among healthcare professionals during future public health crises.

Introduction

The coronavirus was initially reported in Wuhan City, China, in December 2019, which has spread worldwide (World Health Organization, 2020). In terms of COVID-19 protection and management, healthcare professionals are at the forefront. The stressors related to COVID-19 pandemic, coupled with the fear of infection and the emotional toll of witnessing suffering and loss among patients, contribute to detrimental effects on mental health. Due to their duty they are vulnerable to COVID-19, and the high contact rate with COVID-19 patients result in stress and depression during the pandemic (Asnakew, Amha & Kassew, 2021; Shaukat, Mansoor Ali & Razzak, 2020).

The COVID-19 pandemic has had a profound impact on healthcare professionals worldwide, including those in Africa. The healthcare workforce in Africa has long been under-resourced and overburdened, with high levels of burnout and mental health issues already prevalent before the pandemic. The added stress and pressure of the COVID-19 pandemic have likely exacerbated these issues, leading to higher rates of depression, anxiety, stress, and insomnia among healthcare professionals. The ongoing COVID-19 pandemic has affected healthcare systems everywhere, particularly in Africa, with challenges that have never been seen before. The continent has a fragile healthcare infrastructure and limited resources to combat the pandemic’s effects, leading healthcare professionals to face significant physical, emotional, and psychological strains. COVID-19 pandemic and related restrictions can impact mental health (Dragioti et al., 2022). The fear, worry, and uncertainty surrounding such an unknown threat and containment strategies can put a burden on mental health (Chen et al., 2021). A significant number of HCWs manifest various psychological problems in Africa. According to Olashore et al. (2021) reported the psychological impact of COVID-19 on health care workers in African countries and the most common psychological problems were anxiety disorder (with rates ranging from 9.5% to 73.3%) and depression (with rates ranging from 12.5% to 71.9%). The pandemic impacts African healthcare workers through insomnia, depression, and fear of safety in their families (Nchasi et al., 2022).

In Africa until March 2024, about 12,381,551 COVID-19 confirmed cases and 259,261 death were reported by Africa CDC (2024). Africa, as a continent, presents a diverse range of challenges for healthcare professionals. Many countries in Africa struggle with limited resources, inadequate infrastructure, high patient-to-staff ratios, and a lack of mental health support systems. These challenges often place a significant burden on healthcare professionals, leading to increased stress levels, burnout, and ultimately, the development of mental health problems. Likewise, the tension of the pandemic on healthcare professionals cannot be understated. With an unprecedented number of patients, a shortage of necessary medical supplies, and major protocol changes, healthcare workers are being pushed to the brink of their limits. Many are forced to work long hours and take on significantly more patients than they would normally be expected to handle. Moreover, the fact that the threat of COVID-19 lingers has illuminated the higher risk that many healthcare professionals face. To address the mental health crisis amongst healthcare professionals, it is important for employers and healthcare organizations to offer their staff more support. This support can come in the form of offering additional time off, providing counseling services and emotional support, or opening up channels for communication to ensure that healthcare professionals feel heard (Hasen, Seid & Mohammed, 2023c).

During the pandemic, healthcare workers are affected by mental disorders (Hill et al., 2022). The higher prevalence rates of depression in the African population compared to elsewhere (Chen et al., 2021). South Africa’s healthcare professionals working in a tertiary hospital were with high levels of mental health problems during the early COVID-19, and the prevalence of mental distress was 57.4% (Lee et al., 2022). During COVID-19, the prevalence of mental distress was 92.7% among Ugandan health workers (Kirabira et al., 2022). Also in Egypt, HCWs had a high prevalence of anxiety, at 49.38%, insomnia at 56.17%, and poor sleep quality at 67.9% (Al-Otaibi et al., 2022). During the pandemic studies in Ethiopia presented that the prevalence of stress on healthcare professionals was 78.3% (Yitayih et al., 2021), 61.8% (Teshome et al., 2020), 63.7% (Asnakew, Amha & Kassew, 2021), 40.2% (Hajure et al., 2021), 42% (Tsehay, Belete & Necho, 2020), 51.6% (Chekole et al., 2020), 31.4% (GebreEyesus et al., 2021), study in Egypt 66.6% (El-qushayri et al., 2021). and 55.1% (Asnakew, Amha & Kassew, 2021).

Likewise studies in Ethiopia reported that the prevalence of insomnia was 15.9% (Jemal et al., 2021), 50.20% (Yitayih et al., 2021) and 40.8% (Habtamu et al., 2021; Zhang et al., 2021) and global study prevalence was 34.4% (Batra et al., 2020), in Egypt 71.8% (El-qushayri et al., 2021) 73.3% (Sagaon-Teyssier et al. 2020), in Kenya 36% (Kwobah et al., 2021). A global meta-analysis result on the prevalence of anxiety among HCWs found that it was 23.2%, depression 22.8%, and insomnia 38.9% (Rezaei-Hachesu et al., 2022). Besides, a review of studies found that the worldwide prevalence of mental disorders among healthcare workers was between 29.9% and 32.7% had anxiety, 28.4% to 31.3% had depression, and about 40% had sleeping problems (Dragioti et al., 2020). The study findings in the prevalence of mental disorders among healthcare workers with wide disparity evident during this pandemic in Africa (Fernandez et al., 2021). Yet, little reviews were done in Africa regarding the prevalence of mental health problems among healthcare professionals during COVID-19.

Hence, a pooled summary of the results of the meta-analyses on the prevalence of mental health problems among African healthcare professionals during COVID-19 is crucial. This research intends to provide thorough data on the impact of the COVID-19 pandemic on the mental health of healthcare professionals in Africa. Studying an umbrella review on these mental health problems among healthcare professionals in Africa during COVID-19 can provide valuable insights into the magnitude of the problem, the risk factors involved, and potential interventions that can help support the mental well-being of healthcare workers in the region. This research can inform policies and practices aimed at mitigating the impact of the pandemic on the mental health of healthcare professionals and ultimately improving their overall well-being and ability to provide quality care to patients.

Objective

The main goal of this study is to present thorough data on the prevalence of mental health problems among healthcare professionals through COVID-19 in Africa.

Materials and Methods

Protocol registration

We performed an umbrella review based on the Preferred Reporting Items for Systematic Reviews and Meta-Analysis (PRISMA 2020) guidelines for reporting of systematic reviews and meta-analyses presented in Fig. 1. The protocol was registered in the International Prospective Register of Systematic Reviews with PROSPERO registration number: CRD42022383939.

Figure 1 Preferred reporting items for systematic reviews and meta-analysis (PRISMA) flow diagram.

Search strategy

PubMed, African Journal Online, MedRxiv and Google Scholar databases were searched and articles published from the occurrence of COVID-19 from December 2019 to March 2023 were included. Studies from systematic reviews and meta-analyses were taken into consideration to measure the pooled prevalence mental health problems among healthcare professionals in Africa. Every possible keyword combinations were included in systematic searches. Mendeley removed duplicates from the search results (Kwon et al., 2015). The following search terms were used: “COVID-19”, “COVID-19 virus”, “2019 novel corona virus disease”, “SARS CoV-2 infection”, “COVID-19 pandemic”, “COVID-19 infection”, “mental disorder”, “mental illness”, “anxiety”, “depression”, “sleep quality”, “insomnia”, “health workers”, “healthcare professionals”, “nurses”, “doctors”, “pharmacists”, “Ethiopia”, “Algeria”, “Angola”, “Benin”, “Botswana”, “Burkina Faso””, Burundi”, “Cabo Verde”, “Cameroon”, “Central African Republic (CAR)”, “Chad”, “Comoros”, “Congo”, “Democratic Republic of the Congo”, ”Republic of the Cote d’Ivoire”, “Djibouti”, “Egypt”, “Equatorial Guinea”, “Eritrea”, “Eswatini”, “Gabon”, “Gambia”, “Ghana”, “Guinea”, “Guinea-Bissau”, “Kenya”, “Lesotho”, “Liberia”, “Libya”, “Madagascar”, “Malawi”, “Mali”, “Mauritania”, “Mauritius”, “Morocco”, “Mozambique”, “Namibia”, “Nigeria”, “Nigeria”, “Rwanda”, “Sao Tome and Principe”, “Senegal”, “Seychelles”, “Sierra Leone”, “Somalia”, “South Africa”, “South Sudan”, “Sudan”, “Tanzania”, “Togo”, “Tunisia”, “Uganda”, “ Zambia”, “Zimbabwe” (Hasen, Seid & Mohammed, 2023c). Two researchers (AAH and AAS) have screened the titles and abstracts of the studies independently, and any differences between the researchers were resolved by consensus or by the third author (AAM). The search strategy of database is presented in Table 1.

Table 1 The search strategy of databases.

Search number	Search detail	
#1	“COVID-19”[MeSH Terms]	
#2	“mental illness”[Mesh Terms]	
#3	“COVID-19”[Title/Abstract] OR COVID-19 infection ”[Title/Abstract] OR “SARS Cov-2 infection Title/Abstract] OR “COVID-19 pandemic”[Title/Abstract] OR “new corona virus” [Title/Abstract] AND “health care professionals”[Title/Abstract] OR “health workers”[Title/Abstract] OR “nurses”[Title/Abstract] OR “doctors”[Title/Abstract] OR “pharmacists”[Title/Abstract] OR “health care workers”[Title/Abstract] AND “Ethiopia” [Title/Abstract] OR “Algeria” [Title/Abstract] OR“Angola” OR “Benin” [Title/Abstract] OR “Botswana” [Title/Abstract]OR“Burkina Faso” [Title/Abstract]OR “Burundi” [Title/Abstract]OR “Cabo Verde” [Title/Abstract] OR“Cameroon” [Title/Abstract] OR “Central African Republic (CAR)” [Title/Abstract] OR“Chad” [Title/Abstract] OR “Comoros” [Title/Abstract] OR “Congo” [Title/Abstract] OR“Democratic Republic of the Congo” [Title/Abstract] OR”Republic of the Cote d’Ivoire” [Title/Abstract] OR “Djibouti” [Title/Abstract] OR“Egypt”[Title/Abstract], OR“East Africa”[Title/Abstract] OR“South Africa”[Title/Abstract]OR“North Africa”[Title/Abstract] OR“East Africa”[Title/Abstract] OR “Centeral Aftica”[Title/Abstract] OR“West Africa”[Title/Abstract]	
#4	“mental illness” [Title/Abstract] OR “psychiatric problem” [Title/Abstract] OR “mental disorders” [Title/Abstract] AND “anxiety” [Title/Abstract] OR “depression” [Title/Abstract] OR “insomnia” [Title/Abstract] OR “stress” [Title/Abstract] OR “psychology problem” [Title/Abstract] OR “mental health effect” [Title/Abstract] OR “psychological disturbance” [Title/Abstract] AND “meta-analysis” [Title/Abstract] OR “systematic review” [Title/Abstract]	
#5	#1 AND #2	
#6	#3 AND #5	
#7	#4 AND #6	
#8	Limit to systematic review OR meta analysis	

Eligibility criteria

Inclusion criteria

For this umbrella review all systematic review and meta-analysis studies that focused on healthcare professionals and investigations on the impacts of COVID-19 on the mental health of healthcare professionals in Africa were included. This study is employed following the condition, context and population (CoCoPop) framework for the organization of search terms.

Condition: Mental health problems.

Context: Studies conducted in Africa during the COVID-19 pandemic.

Population: This study includes systematic review and meta-analysis studies involving healthcare professionals as a whole.

Study design: Systematic review and meta-analysis studies that report on the prevalence of mental health problems among healthcare professionals during COVID-19 in Africa.

Language: Only studies reported in English.

Publication year: Studies published until March, 2023.

Exclusion criteria

The following studies were excluded: single-level studies i.e., observational studies, studies that included whole population, descriptive reviews, randomized controlled trials, editorials, comments, conference abstracts and expert opinions were excluded.

Outcome measures

The pooled prevalence of mental health problems such as anxiety, depression, stress and insomnia among health care professionals during the COVID-19 pandemic in Africa is the primary outcome in this umbrella review.

Selection of studies

Two researchers (AAH and AAS) measured the studies based on inclusion and exclusion criteria. Firstly, they evaluated both the titles and abstracts of the studies identified from the searched databases. Then full-text screening was done to screen the full texts selected in the previous stage. Besides, we have a justification for the inclusion and exclusion of studies in the PRISMA flow diagram. Finally, the lists of eligible studies for this umbrella review were prepared.

Methodological quality assessment

Two researchers (AAH and AAS) assessed the quality of the included studies by Joanna Briggs Institute’s (JBI) critical appraisal checklist for systematic reviews and research syntheses (Perera et al., 2021). This method assessed the following items: (1) Is the review question clearly and explicitly stated? (2) Were the inclusion criteria appropriate for the review question? (3) Was the search strategy appropriate? (4) Were the sources and resources used to search for studies adequate? (5) Were the criteria for appraising studies appropriate? (6) Was critical appraisal conducted by two or more reviewers independently? (7) Were there methods to minimize errors in data extraction? (8) Were the methods used to combine studies appropriate? (9) Was the likelihood of publication bias assessed? (10) Were recommendations for policy and/or practice supported by the reported data? (11) Were the specific directives for new research appropriate? , and the question in the checklist were answered as “yes”, “no”, “unclear” and Not applicable “NA”(Aromataris et al., 2017). Each entry was evaluated with “yes” or “no”, and the number of “yes” will be counted. Studies score higher than 70% are considered high quality, between 50% and 70% are medium quality and those with a score less than 50% are considered low quality (Hou et al., 2017). Studies with medium and above quality score were considered for analysis.

Data extraction

Two researchers (AAH and AAS) screened the titles and abstracts of all identified articles for eligibility. The required data for this study was extracted by piloted data collection format. After initially screening articles for inclusion based on titles and abstracts, full-text articles were screened. Differences were resolved by deep discussion with a researcher (AAM) to reach an agreement. Data were collected from the selected studies containing, the author’s name, year of publication, country, study design, study population, quality assessment method, searched databases, search restrictions, number of studies, number of reviewers, type of mental illnesses includes depression, anxiety, stress and insomnia, instrument pooled prevalence of mental illness with corresponding 95% confidence interval (CI) were collected.

Data synthesis

Stata version 14.0 (StataCorp, College Station, TX, USA) software was used to conduct this umbrella review. We have calculated pooled prevalence for each mental health problem along with a 95% CI and corresponding p-value. Heterogeneity among eligible study’s findings was assessed using the I2 test, and if I2 >50%, it is considered as a significant heterogeneity (Feilong Zhu et al., 2020). To pool the collected data for each outcome with a 95% CI, DerSimonian-Laird random effects meta-analysis was used.The systematic review characteristics are presented in a tabular form and descriptively. Results of the pooled prevalence of anxiety, depression, stress and insomnia are presented as % with 95% CIs and are graphically presented in the forest plot.

Results

This umbrella review summarized the meta-analysis study findings in Africa on the prevalence of mental health problems among healthcare professionals during the COVID-19 pandemic. Figure 1 shows a PRISMA diagram of the steps in the database search and refinement process for the study of healthcare professionals’ mental health issues during the COVID-19 pandemic. Initial results from our database search included 18 studies. Six studies were removed after 15 studies’ titles and abstracts were examined and three duplicates were removed. There were nine full-text studies examined, and four were excluded for reasons that did not meet the inclusion criteria i.e. single-level studies and studies not in Africa. At long last, we perceived five studies proper to this umbrella review.

Study characteristics

In this umbrella review, we included five systematic review and meta-analysis studies (Rezaei et al., 2022; Hasen, Seid & Mohammed, 2023a; Hasen, Seid & Mohammed, 2023b; El-qushayri et al., 2021; Chen et al., 2021) focused on the impact of COVID-19 on the mental health of healthcare professionals in Africa. The selected meta-analyses were published between 2021 and 2023. When we see the regional/country distribution, one study (El-qushayri et al., 2021) was from Egypt, two studies (Hasen, Seid & Mohammed, 2023a; Hasen, Seid & Mohammed, 2023b) were from Ethiopia, two studies (Rezaei et al., 2022; Chen et al., 2021) were from whole Africa. Depending on the types of mental disorders three studies (Hasen, Seid & Mohammed, 2023a; El-qushayri et al., 2021; Chen et al., 2021) were reported anxiety, four studies (Rezaei et al., 2022; Hasen, Seid & Mohammed, 2023b; El-qushayri et al., 2021; Chen et al., 2021) were reported about depression, two studies (Hasen, Seid & Mohammed, 2023a; El-qushayri et al., 2021) were reported about stress and three studies (Hasen, Seid & Mohammed, 2023b; El-qushayri et al., 2021; Chen et al., 2021) were reported about insomnia.

Regarding the quality appraisal instruments used in the included reviews; three studies used the Newcastle–Ottawa Scale (Rezaei et al., 2022; Hasen, Seid & Mohammed, 2023a; Hasen, Seid & Mohammed, 2023b), one study used the National institute of health quality assessment tool (El-qushayri et al., 2021) and one study applied Mixed Methods Appraisal Tool (Chen et al., 2021). Publication bias was reported in three studies (Hasen, Seid & Mohammed, 2023a; Rezaei et al., 2022; Chen et al., 2021) and two studies (El-qushayri et al., 2021; Hasen, Seid & Mohammed, 2023b) do not report it. In addition, Table 2 contains a summary of the key characteristics of the included studies.

Table 2 Key characteristics of included studies for umbrella review of mental health problems among healthcare professionals during COVID-19 in Africa.

No	Authors	Year	Region/ country	Mental illness	Study Population	Study design	Quality assessment	Searched databases	Search restrictions	# studies	# reviewers	Pooled prevalence %	95% CI	I2 %	Pubcation bias	
1	El-qushayri1 et al.	2021	Egypt	Anxiety	HCWS	SR & MA	National institute of health quality assessment tool	Google Scholar, Scopus, The System For Information on Grey Literature in Europe, Pubmed, The New York Academy of Medicine, and Web of Science	30 December 2020 to 15 January 2021	4	7	72	49.4–86.9	NR	NR	
				Depression	HCWS	SR & MA	National institute of health quality assessment tool	Google Scholar, Scopus, The System For Information on Grey Literature in Europe, Pubmed, The New York Academy of Medicine, and Web of Science	30 December 2020 to 15 January 2021	5	7	65.50	46.9–80.3	98	NR	
				Stress	HCWS	SR & MA	National institute of health quality assessment tool	Google Scholar, Scopus, The System For Information on Grey Literature in Europe, Pubmed, The New York Academy of Medicine, and Web of Science	30 December 2020 to 15 January 2021	7	7	66.60	47.6–81.3	NR	NR	
				Insomnia	HCWS	SR & MA	National institute of health quality assessment tool	Google Scholar, Scopus, The System For Information on Grey Literature in Europe, Pubmed, The New York Academy of Medicine, and Web of Science	30 December 2020 to 15 January 2021	2	7	57.90	45.9–69.0	NR	NR	
2	Hasen et al. A	2023	Ethiopia	Anxiety	HCWS	SR & MA	NOS	Pubmed, Cochrane Library, Crossref and Google Scholar	From the occurrence of COVID-19 pademic to June 2022	8	3	46	30–61	99.07	No	
				Stress	HCWS	SR & MA	NOS	Pubmed, Cochrane Library, Crossref and Google Scholar	From the occurrence of COVID-19 pademic to June 2022	9	3	51	41–62	97.85	No	
3	Hasen et al. B	2023	Ethiopia	Depression	HCWS	SR & MA	NOS	Pubmed, Cochrane Library, Crossref, African Journals Online And Google Scholar	From the occurrence of COVID-19 pademic to June 2022	7	3	40	23–57	99	NR	
				Insomnia	HCWS	SR & MA	NOS	Pubmed, Cochrane Library, Crossref, African Journals Online And Google Scholar	From the occurrence of COVID-19 pademic to June 2022	3	3	35	13–58	98.2	NR	
4	Rezaei et al.	2022	Africa	Depression	HCWS	SR & MA	NOS	Pubmed, EMBASE, Scopus, And Web of Science	January 2019 to February 2021	24	15	82	35–97	NR	No	
5	Chen et al.	2021	Africa	Anxiety	FLHCWS	SR & MA	MMAT	Pubmed, EMBASE, Web of Science, PsycINFO, and medRxiv	1 February 2020 to 6 February 2021	3	12	51	31-70	NR	No	
				Depression	FLHCWS	SR & MA	MMAT	Pubmed, EMBASE, Web of Science, PsycINFO, and medRxiv	1 February 2020 to 6 February 2021	3	12	55	32-76	NR	No	
				Insomnia	FLHCWS	SR & MA	MMAT	Pubmed, EMBASE, Web of Science, PsycINFO, and medRxiv	1 February 2020 to 6 February 2021	3	12	30	2–71	NR	No	
				Anxiety	GHCWS	SR & MA	MMAT	Pubmed, EMBASE, Web of Science, PsycINFO, and medRxiv	1 February 2020 to 6 February 2021	12	12	35	23–48	NR	No	
				Depression	GHCWS	SR & MA	MMAT	Pubmed, EMBASE, Web of Science, PsycINFO, and medRxiv	1 February 2020 to 6 February 2021	12	12	43	39–58	NR	No	
				Insomnia	GHCWS	SR & MA	MMAT	Pubmed, EMBASE, Web of Science, PsycINFO, and medRxiv	1 February 2020 to 6 February 2021	12	12	42	31–55	NR	No	
Notes.

SR & MA Systematic review and meta-analysis

MMAT Mixed methods appraisal tool

NR not reported

NOS Newcastle Ottawa Quality Assessment Scale

HCWS healthcare workers

FHCWS frontline healthcare workers

GHCWS General healthcare workers

See El-qushayri et al. (2021); Hasen, Seid & Mohammed (2023a); Hasen, Seid & Mohammed (2023b); Rezaei et al. (2022); Chen et al. (2021).

Quality of included studies

A quality score for the five included studies is shown in Table 3 using the Joanna Briggs Institute (JBI) critical appraisal checklist for systematic reviews and research syntheses. Accordingly, all studies (Rezaei et al., 2022; Hasen, Seid & Mohammed, 2023a; El-qushayri et al., 2021; Chen et al., 2021; Hasen, Seid & Mohammed, 2023b) were considered as high quality based on their JBI quality score higher than 70%, and thus considered for final umbrella review.

Table 3 Joanna Briggs Institute (JBI) critical appraisal checklist for systematic reviews and research syntheses.

No	Study(year)	Q1	Q2	Q3	Q4	Q5	Q6	Q7	Q8	Q9	Q10	Q11	Total (%)	Overall appraisal	
1	El-qushayri et al. (2021)	Y	Y	Y	Y	Y	Y	N	Y	N	Y	Y	9/11=81.81%	Include	
2	Hasen, Alebachew Seid & Adem Mohammed (2023a)	Y	Y	Y	Y	Y	Y	Y	Y	Y	Y	Y	11/11=100%	Include	
3	Hasen, Alebachew Seid & Adem Mohammed (2023b)	Y	Y	Y	Y	Y	Y	Y	Y	N	Y	Y	10/11=90.90%	Include	
4	Rezaei et al. (2022)	Y	Y	Y	Y	Y	Y	Y	Y	Y	Y	Y	11/11=100%	Include	
5	Chen et al. (2021)	Y	Y	Y	N	Y	Y	Y	Y	Y	Y	Y	10/11=90.90%	Include	
Notes.

Y Yes

N No

U Unclear

NA Not applicable

Study overlap assessment

To assess the study overlap we used the calculation of the overall corrected covered area (CCA) approach (Pieper et al., 2014). Accordingly we assessed the study overlap in studies of stress, anxiety, deprssion and insomnia. The CCA is reported as for stress CCA =0%, anxiety CCA =4.5%, depression CCA =3.3% and Insomnia CCA =11.1% a shown in Fig. S1. We found a high levelof overlap in insomia studies.

The pooled prevalence of depression, anxiety, stress, and insomnia

As shown in the forest plot (Fig. 2), the prevalence of common mental health problems (depression, anxiety, stress and insomnia) among healthcare professionals during COVID-19 in Africa was found to be high. Accordingly, a total of five studies reported the prevalence of depression, and the pooled prevalence of depression was found to be 53.75% (95% CI [40.80–66.70], I2 = 63.6%, p = 0.027), and I2 = 63.6%, indicating that there exists significant heterogeneity among study findings on the prevalence of depression among health care professionals through the pandemic in Africa. Similarly, four studies reported the prevalence of anxiety, and the pooled prevalence of anxiety was found to be 49.97% (95% CI [34.71–65.23], I2 = 71.26%, p = 0.014). There exists significant heterogeneity among study findings on the prevalence of anxiety among healthcare professionals in Africa through the pandemic I2 = 71.26%. This study also reported the pooled prevalence of stress, and from a total of 2 studies, the pooled prevalence of stress was 57.27% (95% CI [42.28–72.25], I2 = 58.9%, p = 0.119). No statistically significant heterogeneity of study results has been observed; this might be due to the limited studies included in the meta-analysis. Likewise, from a total of four studies, the pooled prevalence of insomnia was 45.16% (95% CI [32.94–57.39], I2 = 50.8%, p = 0.107), and no statistically significant heterogeneity of study results has been observed.

Figure 2 Forestplot of pooled prevalence of mental disorders among healthcare professionals during COVID-19 in Africa.

Discussion

The current comprehensive umbrella review aimed to determine the pooled prevalence of mental health problems during the COVID-19 pandemic among healthcare professionals in Africa. This review detected five systematic reviews and meta-analyses that evaluated the prevalence of mental health symptoms among healthcare professionals during the COVID-19 pandemic. From the included meta-analysis there is heterogeneity among studies. The variations might be: For example, study populations can greatly impact heterogeneity in research findings. Different demographics, socioeconomic backgrounds, cultural norms, and healthcare access among study participants can all influence how interventions are received and implemented, leading to varying outcomes. Methodologies are another critical factor that can contribute to heterogeneity in research. Variations in study design, data collection methods, sample sizes, and statistical analysis techniques can all affect the reliability and validity of study findings. For instance, using different measurement tools or data collection techniques can produce inconsistent results, making it difficult to compare or generalize findings across studies. However, it is important to consider the implications of heterogeneity on the validity of the study findings. High levels of heterogeneity may indicate that the results are not consistent across studies, which could affect the generalizability of the findings.

The most commonly studied symptoms across all meta-analyses were depression, anxiety, stress and insomnia. The pooled prevalence of depression, anxiety, stress and insomnia is discussed as follows. This umbrella review demonstrates the prevalence rates for depression among healthcare professionals during COVID-19 in Africa, and the pooled prevalence of depression was found to be 53.75% ranging from 40.80% to 66.70%. This is higher than the meta-analysis findings in China, where the pooled prevalence of depression was 39% (Xiong et al., 2022) and also worldwide depression pooled prevalence of 13.4% (Hill et al., 2022), 37% (Saragih et al., 2021), 22.8% (Rezaei-Hachesu et al., 2022). Also the result is higher than globally reported umbrella review prevalence of depression was 28.4% (Dragioti et al., 2020), 17.9% (Fernandez et al., 2021). Compared to a studies in Asia, with pooled prevalence of depression 37.5% (Ching et al., 2021), 34.61% (Norhayati, Che Yusof & Yacob Azman, 2021), the magnitude is high in africa. The cause for the high prevalence and differences of depression in Africa among healthcare professionals during COVID-19 might be due to there were growing concerns over the challenges on the African healthcare workers such as economic insecurity and stressful working conditions, limited access to personal protective equipment and other vital resources such as ventilators. Moreover, the pandemic influences African healthcare workers to social stigma, burnout, insomnia, depression, and fear of safety in their families (Nchasi et al., 2022).

Similarly, the pooled prevalence of anxiety was found 49.97% ranged from 34.71% to 65.23%. This is higher than meta-analysis findings in China pooled prevalence of anxiety was 37% (Xiong et al., 2022) and also worldwide anxiety pooled prevalence 16.1% (Hill et al., 2022), 40% (Saragih et al., 2021). The result is higher than global umbrella review results prevalence of anxiety 29.9% (Dragioti et al., 2020), 22.2% to 33.0% (Fernandez et al., 2021). It is higher than the study in Asia reported the pooled prevalence of anxiety 39.7(95%) (Ching et al., 2021), 34.81% (Norhayati, Che Yusof & Yacob Azman, 2021).

The pooled prevalence of stress was 57.27% ranged from 42.28% to 72.25%. This is much higher than meta-analysis findings in worldwide pooled prevalence of stress was 37% (Saragih et al., 2021), Asia 36.4% (Ching et al., 2021), 31.72% (Norhayati, Che Yusof & Yacob Azman, 2021).

Our study also reported the prevalence of insomnia among African healthcare professionals during the COVID-19 pandemic, and the pooled prevalence of insomnia was 45.16% ranged from 32.94% to 57.39%. Compared to the global findings, it is high in magnitude. For instance, a pooled prevalence of insomnia was estimated at 38 9% (Rezaei-Hachesu et al., 2022). The result is higher than the global umbrella review result prevalence of insomnia was 40% (Dragioti et al., 2020), Asia 37.89% (Norhayati, Che Yusof & Yacob Azman, 2021). The high heterogeneity in the pooled prevalence of mental health problems in Africa compared to others could be due to a variety of factors, including: Socioeconomic factors: Africa has a diverse range of socio-economic conditions and disparities, which could contribute to differences in the prevalence of anxiety disorders across different regions within the continent. Access to mental health services: Limited access to mental health services and treatment options in many African countries could result in underreporting of anxiety disorders and higher levels of untreated anxiety. Methodological differences: Variations in study design, methodology, and sampling techniques across different studies could also contribute to the heterogeneity in the pooled prevalence of anxiety in Africa compared to other.

This umbrella review helps primarily to assess the magnitude of mental health problems among healthcare professionals during COVID-19 in Africa. Literatures have indicated Africa as psychologically vulnerable during the pandemic, however such reports are often not summarized, and our umbrella review tried to indicate the mental health problems such as depression, anxiety, stress, and insomnia were highly prevalent in Africa. The study’s findings highlight the need for increased awareness and interventions to address mental health problems among healthcare professionals in Africa during the COVID-19 pandemic. This could include implementing mental health support programs, providing training on coping strategies, and developing policies to support mental health needs. The study’s findings have implications for policy development around mental health support for healthcare professionals in Africa. Policymakers may need to consider implementing guidelines and regulations to support mental health needs, as well as incorporating mental health support into healthcare policies and systems. Also this study involved on the importance of collaboration and partnerships in addressing mental health problems among healthcare professionals in Africa. This could involve collaboration between healthcare providers, government agencies, mental health organizations, and other stakeholders to develop and implement comprehensive mental health support programs. Also our umbrella review highlights the need to improve the quality of systematic reviews and meta analysis in this field of research. Additionally, summarizing existing meta-analyses can help identify gaps and inconsistencies in the existing literature, which can guide future research efforts. By synthesizing the findings of multiple meta-analyses, the researchers can identify common themes and trends in the existing evidence on mental health in Africa, as well as areas where further research is needed.

Whereas, this study has strengths and limitation. To our knowledge, this is the first umbrella review to examine the pooled prevalence of depression, anxiety, stress and insomnia among healthcare professionals during COVID-19 in Africa. One strength of this methodology is that it offers comprehensive evidence on the mental health challenges faced by healthcare professionals during the COVID-19 pandemic in Africa. The screening, data extraction and methodological quality assessment were done by two authors independently. The quality assessment was checked by the Joanna Briggs Institute (JBI) critical appraisal checklist for systematic reviews and research syntheses. Moreover, this study presents the predictive interval in which future studies finding might be in it. However, there are potential limitations to be aware of. Firstly in the absence of sufficient meta-analysis studies on the prevalence of mental health problems among healthcare professionals in Africa, subgroup analyses were not performed due to the limited studies included. Secondly, there is a risk of selection bias as non-English language meta-analyses were not included, potentially skewing the results. Thirdly, there is overlap in primary studies across meta-analyses of insomnia studies might leading to biased estimates. Fourthly, Hetrogeneity in methodologies among primary studies, including sampling methods and assessment tools, could impact the accuracy of prevalence estimates for stress, anxiety, depression, and insomnia. Lastly, it is important to recognize that these mental health problems may vary among different groups of healthcare professionals, highlighting the need for future research to explore prevalence estimates within specific circumstances.

Conclusions

In conclusion, healthcare professionals in Africa were at a high risk of experiencing a variety of mental health problems during the COVID-19 pandemic. The prevalence of anxiety, depression, stress and insomnia among healthcare professionals were significantly higher in Africa compared to other regions. This study’s findings highlight the need to develop appropriate public health interventions to address healthcare professionals’ mental health needs in Africa during COVID-19 and related public health crises.

Supplemental Information

Supplemental Information 1 PRISMA checklist

Supplemental Information 2 The rationale for conducting the Umbrella review

Figure S3 Graphical presentation of assessment of study overlap in reviews using Corrected Covered Area (CCA) method

Supplementary figure 1

Abbreviations

CI Confidence Interval

HCWs Health Care Workers

JBI Joanna Briggs Institute

MeSH Medical Subject Headings

PRISMA Preferred Reporting Items for Systematic Review and Meta-Analysis

Additional Information and Declarations

Competing Interests

Author Contributions

Data Availability

The authors declare there are no competing interests.

Aragaw Asfaw Hasen conceived and designed the experiments, performed the experiments, analyzed the data, prepared figures and/or tables, authored or reviewed drafts of the article, and approved the final draft.

Ahmed Adem Mohammed conceived and designed the experiments, performed the experiments, analyzed the data, prepared figures and/or tables, and approved the final draft.

Abubeker Alebachew Seid conceived and designed the experiments, performed the experiments, analyzed the data, prepared figures and/or tables, authored or reviewed drafts of the article, and approved the final draft.

The following information was supplied regarding data availability:

All data relevant to the study are included in the article.

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
