# Peer review of "Prevalence of anxiety, depression, stress and insomnia among healthcare professionals during COVID-19 in Africa: umbrella review of existing meta-analyses"

_PeerJ, doi:10.7717/peerj.18108_

## Round 0.1 · original submission · Minor Revisions

Please address the concerns of all reviewers even though the editorial verdict is that of minor revision in view of the correctness of the methods. However, in your return response, please justify your choice of analysis (meta-review as opposed to a new meta-analysis of included studies), address all grammatical and typographical errors that the reviewers have pointed out, and tidy up the paper.

**Language Note:** The Academic Editor has identified that the English language must be improved. PeerJ can provide language editing services - please contact us at [email protected] for pricing (be sure to provide your manuscript number and title). Alternatively, you should make your own arrangements to improve the language quality and provide details in your response letter. – PeerJ Staff

Reviewer 1 ·

Basic reporting

You should check typos
You should check your references in the text and references section.

Experimental design

no comment

Validity of the findings

no comment

Annotated reviews are not available for download in order to protect the identity of reviewers who chose to remain anonymous.

Reviewer 2 ·

Basic reporting

Please find additional comments.

Experimental design

Please find additional comments.

Validity of the findings

Please find additional comments.

Additional comments

ID: peerj-96252-1
Title: Prevalence of anxiety, depression, stress and insomnia among healthcare professionals during COVID-19 in Africa: umbrella review of existing meta-analyses

Thank you for providing a chance to review this manuscript.

I finished all the reading. Numerous grammar errors occur, indicating the language should be improved throughout the manuscript. Although the authors aim to summarize mental health issues among African healthcare workers during COVID-19, there are several concerns with the study. My comments are as follows:
Keywords
Overall: Where are your keywords set? Keywords are an important part of an article whether it is read or retrieved, and authors are advised to add them.
Abstract
Overall: Please add page numbers to your post to pinpoint the problem.
Line 30-31, Page 6: Since this is an “Umbrella review of existing meta-analyses”, the background in the Abstract should also refer to the situation of literature on the mental health problems of healthcare professionals in Africa.
Line 29, Page 6: “Mental health problems”, a small suggestion, it would be to directly refer to “anxiety, depression, stress, and insomnia”.
Line 34-40 Page 6: Personally, I think that such a methodological presentation in the Abstract is a bit too redundant, e.g. the statistical software used does not need to be listed.
Line 41-46, Page 6: Be careful with italics, e.g. “p” needs to be italicized.
INTRODUCTION
Line 58-60, Page 7: Can you show detailed data, phenomena or research about the high workloads, equipment shortages and higher risk of exposure to the virus of healthcare professionals on the frontlines to support this argument?
Line 56-65, Page 7: Didn’t see any description of what the prevalence of COVID-19 is like in Africa? Please add.
Line 66-67, Page 7: I’m puzzled by the timeframe outlined in this research background. As of 2024, COVID-19 isn’t globally prevalent but occurs sporadically. So, I suggest adjusting the pandemic’s portrayal to reflect the current situation. Sorry that I couldn’t find updated COVID-19 data for Africa. If available, please incorporate it to ensure the background and context align with the present reality.
Line 67-69, Page 7: There is an argumentative disconnect, how does the current state of resource scarcity in Africa lead to so many facts of strains for healthcare professionals? What specifically are these strains? All relevant elements are suggested to be added.
Line 76-78, Page 7: There is a duplication of content in the literature and the terms “social stigma, burnout, insomnia, depression, and fear” are not clearly relevant to your study. It is not clear why this statement was made at the end of the paragraph.
Paragraph 3, Page 7-8: The paragraph, especially the beginning, demonstrates a central meaning very similar to that of Paragraph 2, and a repetitive statement is not recommended.
Paragraph 2-4, Page 7-9:
1) Could the authors clarify how this study supplements the existing literature on the psychological health issues confronting African healthcare professionals amid the COVID-19 pandemic? What unique contributions does it offer to this domain? Specifically, could you delineate which gaps in prior research it addresses or what novel insights it introduces?
2) Given the extensive research in recent years on the psychological impacts of the pandemic period/post-pandemic period on various populations (with healthcare professionals being a focal group), how did the authors accommodate the complexity and diversity of psychological health challenges experienced by African healthcare professionals during the COVID-19 pandemic? How are these challenges influenced by factors such as the pandemic itself, cultural disparities, and societal dynamics?
3) While copious data can furnish a thorough research backdrop, could the authors expound upon the significance and practical utility of this study? How does the research framework contribute to enhancing the resilience of African healthcare systems and fostering a conducive working environment for healthcare professionals?
4) Based on the current trend, the implementation of vaccination and other control measures may have alleviated the impact of the pandemic on society and individual lives. Therefore, this study seems to be lagging and lacks a comprehensive synthesis of existing research as well as a precise positioning of unresolved issues in the research field.
Line 117-118, Page 9: “Yet, little reviews was done in Africa regarding the prevalence of mental health problems among healthcare professionals during COVID-19”, what is the number of articles investigating relevant research, excluding those conducting reviews? What is the situation or quality? A relevant review is necessary.
Line 119-120, Page 9: The authors lack a reasonable explanation and justification for the choice of research methodology. Why did you choose to conduct a meta-analysis, and what’s the effectiveness of meta-analysis in this research domain? This study merely entails a repetitive summary and analysis of existing research, without offering novel insights or conclusions. Such an approach lacks innovation and originality, failing to contribute new knowledge or address practical issues in the academic field.
Line 123-125, Page 9: Merging in the introduction is sufficient, and it is not necessary to make a separate subheading on how brief it is.
Overall: Both the title and the abstract talk about “anxiety, depression, stress, and insomnia”; however, the abstract does not provide a detailed description of each variable, such as insomnia. The addition of each is necessary.
Overall: Each paragraph of the background introduction does not have a clear and distinguishable, relevant single theme, and much of the narrative content is repeated and argued for in multiple paragraphs, making the entire background narrative seem illogical.
MATERIALS AND METHODS
Protocol registration
Figure 1, Page 25: Some details in the picture need to be improved, such as the need for a space before and after the “=”.
Search strategy
1) The current inclusion of literature seems somewhat limited. Perhaps the authors could consider incorporating more comprehensive medical databases, such as Web of Science, Scopus, Embase, etc.
2) In Table 1 #3, there are grammatical and symbol errors in the logical operations, and the writing is not as neat as in #4. Please revise Table 1 #3 accordingly.
3) Specific search strategies can be moved to the appendix.
Eligibility criteria
1) The inclusion dates of the literature need to be specified accurately down to the day.
2) While the current standards appear relatively clear, there are still some areas that need further clarification. For instance, the authors could provide a more specific definition of the study population, specifying whether it includes all healthcare professionals or is specific to certain professions or job titles. Additionally, the author could consider adding specific requirements regarding study design to ensure that the included studies have a certain methodological quality.
Data extraction: This section could benefit from clearer details on the screening procedure, including how titles and abstracts were reviewed for eligibility, the procedure for full-text article screening, and the resolution of any discrepancies. Adding specifics on the tools used and the timeframe for data extraction would enhance clarity. Additionally, more insights into the quality assessment methods and criteria would improve understanding of how study quality was evaluated. These adjustments would streamline the description of the data extraction process while maintaining comprehensiveness.
DISCUSSION
1) The discussion section should align with the initial study design, yet the authors appear to have included content about stress while neglecting the discussion of insomnia.
2) While the authors have summarized the pooled prevalence rates of various mental health issues and compared them with findings from other regions and globally, adequate explanation is needed to clarify these results’ significance and potential implications.
3) So, what specific factors might have led to the increased occurrence of mental health issues among healthcare workers in Africa during the COVID-19 pandemic? While the authors suggest that their findings could influence policies and practices, how will these potential impacts be put into action? As previously mentioned, what practical significance do the results of this study hold?


I have thoroughly reviewed the entire manuscript, and I must commend the authors for conducting an exhaustive study. The authors have undertaken a substantial amount of work, which is undeniable. Nevertheless, based on what was presented, there are still more issues that prevented this study from being published, among which stand out the lack of a smooth and logical presentation of the background, the lack of innovation demonstrated, and the quality of the literature covered is too low. The author can refer to the remaining high quality journals to make corrections to the article.

Thank you and my best
Your reviewer

Reviewer 3 ·

Basic reporting

This study has provided sufficient field context regarding the need for attention regarding the mental health status among healthcare professionals in Africa. The structure of the article is good. The figures are relevant but need some further improvement in the resolution and labeling. There are several areas in the formatting and detail of reporting that need further attention to meet the publication standards. Below, I have outlined detailed comments to improve the basic reporting of this manuscript:

Line 70-72 Could you please provide a citation for this
Line 94 There is a missed space between “disorders” and “(Hill et al. 2022)”
Line 74 Please spell out HCWs because this is the first time you have used this abbreviation
Line 215 there were two extra spaces between “and if” and “I2>0.5”, and between “p<0.1” and “, it is”
Line 216-217 Should this citation be referred as something like “Zhu et al. 2020”?
Line 218 There is a space between “used.” and “The”
Line 290-322, Figure 2 Please try to present the results using the same precision, i.e. same decimal place.
Line 294 “22.8%” was not using the correct decimal sign
Line 292-295 “This is higher than … pooled prevalence 13.4%, 37%, 22.8%” Please rewrite this sentence with appropriate grammar.
Line 296-298 “It is higher than… 37.5%, 34.61%” Please rewrite this sentence with appropriate grammar. There is a missing space between “17.9%” and “(Fernandez et al. 2021).”
Line 308-309 Should this citation be referred as something like “Saragih et al. 2021”?
Line 314-316 There is an extra space between “worldwide” and “pooled prevalence”, and there has missed a space between “was 37%” and “(Ita Daryanti Saragih, …, 2021)”. Should this citation be referred as something like “Saragih et al. 2021”?
Line 320-322 There is an extra space between “instance” and “a”, and between “of” and “insomnia”. The decimal point was incorrect for 38.9%. Please rewrite the sentence with appropriate grammar. Please include the location information of the findings from Dragioti et al. 2020.
Line 331 There is an extra space between “pandemic.” and “Moreover”
Line 357 “wellbeing” should be corrected into “well-being”
Line 339 It should be “Joanna Briggs Institute (JBI) critical appraisal checklist” instead of “Joanna Briggs Institute JBI critical appraisal checklist”
Line 441 and Line 444 the authors of the two citations were not listed. This seems to be a citation error.
Figure 2
1. The resolution of the figure seems too low.
2. In the x-axis, please remove the label “-100”. There are no negative values in terms of the prevalence of a disease.
3. Please spell out “DL” in the footnote
Table 2
1. Please be consistent with the precision of the number reported, i.e. please use the same decimal place for pooled prevalence and its 95% CI, and use the same decimal place for I2.
2. Please list what is “>>”, “NR”, “CI”, and “I2” means in the footnote.

Experimental design

Research questions are well-defined and relevant. Rigorous investigation performed to a high technical & ethical standard. The author performed a rigorous investigation. And this is a highly technical study in terms of the study methods. Methods are described with sufficient detail & information to replicate. However, there are major concerns with the design mainly regarding the small sample size.
- Major concern
The author needs to provide more evidence regarding the meaningfulness of the research. This study is a summary study of 5 meta-analyses. The author needs to justify why they summarize the 5 existing meta-analyses instead of conducting a new meta-analysis, in this way, there will be less bias introduced. The author also needs to examine if there are overlaps among the studies included in the 5 meta-analyses.
The small sample size might also introduce issues to the I2 estimates from DerSimonian and Laird. Please also consider other methods such as Maximum likelihood estimation [1], Knapp‐Hartung method [2], and/or Kartung-Knapp-Sidik-Jonkman method [3].

Reference
1. Jackson, D., Bowden, J., & Baker, R. (2010). How does the DerSimonian and Laird procedure for random effects meta-analysis compare with its more efficient but harder to compute counterparts?. Journal of Statistical Planning and Inference, 140(4), 961-970.
2. Bender, R., Friede, T., Koch, A., Kuss, O., Schlattmann, P., Schwarzer, G., & Skipka, G. (2018). Methods for evidence synthesis in the case of very few studies. Research synthesis methods, 9(3), 382–392. https://doi.org/10.1002/jrsm.1297
3. IntHout, J., Ioannidis, J.P. & Borm, G.F. The Hartung-Knapp-Sidik-Jonkman method for random effects meta-analysis is straightforward and considerably outperforms the standard DerSimonian-Laird method. BMC Med Res Methodol 14, 25 (2014). https://doi.org/10.1186/1471-2288-14-25

Validity of the findings

All underlying data have been provided; however, there are potential issues (listed below) with the robustness of the analysis. Conclusions are well stated, linked to the original research question & limited to supporting results; however, the conclusion might be impacted by the potential issues stated below.
- Major concern
1. The study sample size is too small. The study only includes 5 meta-analyses and this might raise questions to the validity and generalizability of the findings, especially for stress where there are only 2 studies included. The authors should consider including more recent studies when they define their search strategy, for example, include studies till March 2024 instead of till March 2023 in the article.
2. There is heterogeneity reported for depression and the authors should assess its impact on the validity of the findings of the research.
3. There is also ambiguity regarding the reporting of heterogeneity. You mentioned that results with I2>0.5 or p<0.1 are considered significant heterogeneity. However, according to this standard, your results for depression, anxiety, stress, insomnia, and overall all should be considered as significant heterogeneity. But in the manuscript, it seems that you have used the standard that I2>0.5 AND p<0.1 for heterogeneity.

Additional comments

Line 75-76
You mentioned that from Johnson et al, the anxiety disorder rate ranged from 9.5% to 73.3% and the depression rate ranged from 12.5% to 71.9%. The range of the rates is too high to provide useful information here. And also please provide the location and population of those estimates.
Line 117-120
I appreciate your introduction to the need for studying the mental health problems among African healthcare professionals during COVID-19. However, you did not justify why a pooled summary of the meta-analyses is needed. You should provide your reason and the necessity for a pooled summary of the meta-analyses instead of a meta-analysis of all the existing studies. Moreover, you only involved 5 meta-analyses in this manuscript, please justify the need for a need of an umbrella review of 5 existing meta-analyses.
Also, did you check if the studies included in the 5 meta-analyses have any overlapping studies, i.e. if a single paper was included in multiple meta-analyses? If so, this might introduce uncertainty in the pooled estimate in your study.

Line 134
You mentioned that you searched all the meta-analyses till March 2023. However, it has been more than one year since March 2023, I am wondering if you could update your study with a wider search strategy to include the studies that are published at a later time point, for example, March 2024? And this will also help with the small sample size issue in this study.
Line 215
You mentioned that results with I2>0.5 or p<0.1 are considered significant heterogeneity. However, according to this standard, your results for depression, anxiety, stress, insomnia, and the overall all should be considered as significant heterogeneity. Please describe the impact of the heterogeneities might have to the validity of study results and conclusion.
Please be consistent about the reporting of I2. Here I2 was reported as 0.5, but in the rest of manuscript and the table 2, it was reported as a percentage.
Line 244
You mentioned that 2 studies were from the whole of Africa. Could you please provide more detailed information about where the study is from, for example, be specific about how many counties of Africa are included in the studies.

Line 270
You mentioned that for depression, the I2 = 63.6%, indicating that there are heterogeneity among the study. You should discuss further information about why there are heterogeneity among studies, what information does this heterogeneity tells us, and will this heterogeneity affect the validity of the study finding. Please add those information in the discussion section

Line 292, Line 305-306, Line 313, Line 319
You included statements like “pooled prevalence of depression was found 53.75% ranged from 40.80% to 66.70%” Please double check this because ranged from means the minimal value to the maximum value. It seems that you are referring to the 95% prediction interval in Figure 2. If so, you should present it like “pooled prevalence of depression was found 53.75% (95% Prediction interval: 40.80%-66.70%)”

Line 306-307
“Pooled prevalence of anxiety was 37% (32-42%)”. Is this 32-42% reported as 95% CI? Please specify it in the manuscript

Line 329
I appreciate that you are trying to show a regional difference of the mental health status in Africa. However, it is not clear how this manuscript shows the regional differences as only 5 studies were included, 2 are in Ethiopia, 1 is in Egypt, and 2 are from whole Africa (without the detailed location of the countries included ). Please clarify how your sample is showing the regional differences and maybe consider summarizing the regional differences in the result section.

Line 335-344
There are several limitations that are not included in this paragraph
1. There is a limited number(5) of studies included this manuscript, this might challenge the generalizability of the study results and conclusion
2. This manuscript is a summary of meta-analysis, which could potentially introduce additional bias in the study results.
3. Your results showed a heterogeneity in depression results, this heterogeneity is likely to affect the validity of the study results

Line 352
Please be specific about the prevalence, for example, the prevalence of mental health issues among healthcare professionals.

Figure 2
In the last row, you reported the results for “Overall” is 50.68 (44.47, 56.90). Please specify what this overall stands for. If it stands for combined mental disorders of Depression, Anxiety, Stress, and Insomnia, why this overall estimate is smaller than the prevalence of its single parts, for example, your finding is that the overall prevalence of mental disorder is 50.68%, but the prevalence of depression is 53.75%.
Table 1
For the item #3, you mentioned that you searched the term OR”SouthAfrica”[Title/Abstract]OR”NorthAfrica”[Title/Abstract]OR”EastAfrica”[Title/Abstract]. Do you miss the space between the words? Please specify how this was searched.

Table 2
1. The # of studies was missing for Rezaei et al
2. For the column I2, is this the heterogeneity? If so, please add the percentage sign in the column name. Please also report the p values related to the I2. And also the included studies all have extremely high heterogeneity, ranging from 97.8 to 99. Please discuss why the heterogeneity was so high and if this will affect the validity of the results and conclusion in your manuscript.
3. Razei et al’s search restrictions were from Jan 2019 to Feb 2021. However, COVID-19 started in December 2019, did Razei et al include studies published before COVID-19 started?
4. Chen et al reported 3 studies of frontline healthcare workers and 12 studies of general healthcare workers. Did those studies overlap? Is Chen et al included a total of 12 studies or 3+12=15 studies.
5. What is the difference between healthcare workers, general healthcare workers, and frontline healthcare workers, especially healthcare workers vs general healthcare workers?

Reviewer 4 ·

Basic reporting

The authors present an important and timely umbrella review focusing on the mental health challenges faced by healthcare professionals in Africa during the COVID-19 pandemic. The paper clearly outlines the aim of the study, which is to assess the prevalence of anxiety, depression, stress, and insomnia among healthcare professionals in Africa during the COVID-19 pandemic. The authors utilized systematic searches across multiple databases, ensuring a thorough review of existing literature. The paper presents a valuable contribution to the literature on mental health among healthcare professionals in Africa during the COVID-19 pandemic. Addressing the suggested improvements would strengthen the clarity and reliability of the study's findings.

Experimental design

1. Given that "stress" was highlighted as a primary outcome, further clarification is needed regarding the specifics of the term "stress" used as a search criterion (refer to lines 140-141).

2. Could you specify the exact mental health conditions, such as anxiety, included in the search (refer to line 160)?

3. Regarding the population (lines 162-163), could you define "healthcare professionals"? Did they need to have direct contact, either face-to-face with SARS-CoV-2 infected patients or other patients? If the source articles lacked clear information, what assumptions were made in this manuscript?

4. Could you provide the reference source for the "piloted data collection format" mentioned at line 203?

5. In the "Data extraction" section (line 201), please clarify if all types of mental illnesses were initially considered. Also, specify whether all results aligning with each outcome domain in every study were sought. If not, explain the methods used to determine which results to collect. For instance, how were depression symptoms measured in different studies, utilizing tools like PHQ-2, PHQ-8, and PHQ-9?

6. The reasons for excluding the four articles mentioned in line 235 are unclear. They were selected for review (lines 233-234) based on meeting inclusion criteria. It's expected to provide reasons for how these articles met exclusion criteria.

7. Could you explain why three quality appraisal instruments were used for these articles (lines 250-254), and why specific instruments were chosen for each group of articles? Then, why was the Joanna Briggs Institute (JBI) critical appraisal checklist selected to assess all studies (line 259)? Please provide a reference resource for the JBI checklist.

8. Heterogeneity Assessment: While I2 statistics are provided to assess heterogeneity, further discussion on the potential sources of heterogeneity and their implications for the interpretation of results would enhance the robustness of the analysis.

Validity of the findings

1. Given that prevalence is the main outcome measurement, it's crucial to comment on the sampling approach (probability sampling or convenience sampling) in the original studies. Prevalence estimation from a convenience sample may not accurately represent the broader population due to selection bias. Therefore, prevalence estimates from convenience samples should be interpreted cautiously and may not be generalizable. It's suggested to carefully review the sampling approaches in the source studies and acknowledge any limitations if convenience sampling was used.

2. Despite the eligibility period being between the onset of the pandemic and March 2023, the various waves of the pandemic may result in fluctuations in mental health conditions. Consideration of the comparability of the timing of source data is essential.

3. The discussion section predominantly focuses on comparing the estimations of this manuscript with global or Asian data. However, without details on sampling approaches and timing of assessments, valid comparability is challenging. Instead of framing most of the discussion around comparisons to global or Asian data (lines 290-322), it's suggested to expand the discussion to include factors contributing to the high prevalence, prevalence data of such mental health conditions among the general population in Africa, efforts aimed at reducing these issues among healthcare

---

## Round 0.2 · Minor Revisions

Please address all the comments (minor revisions as well as those made by R3) in view of:

"justify the need for this umbrella review of the existing meta-analyses. While the authors have explained the need for a review of mental health status in Africa, this goal might be better achieved through a meta-analysis of all available studies rather than pooling results from only five existing meta-analyses", consider how analysis of meta analyses might be similar and different from pooling of results from individual studies. This is not a trivial question. Here, check the overlapping studies, and see if you can pool the results of all studies that are not overlapping in your sample and run a meta analysis on the side. It should not be time consuming with most standard meta analysis packages. This will also address the third comment from the reviewer who suggested "rejection".

For the second comment, you may want to state that this review is current as of 2023, and acknowledge that more studies may have been published but this is a snapshot. This is just a suggestion, but please address this criticism with logic.

Given that two reviewers suggested acceptance, one reviewer suggested minor revision, and one reviewer suggested rejection (but the clauses can be addressed), I suggest you do a minor revision of the paper and resubmit. Please address these concerns.

Reviewer 1 ·

Basic reporting

.

Experimental design

.

Validity of the findings

.

Additional comments

The researchers have addressed an important issue. The manuscript meets all the scientific standards necessary for publication. The title effectively conveys the content of the manuscript, and the abstract includes sufficient and necessary data. Further statistical analyses are not required. In fact, the analyses conducted by the researchers are sufficient. The Results and Discussion sections are sufficient and well-organized. The language of the manuscript is suitable for the intended audience and contains only minor spelling and punctuation errors. The manuscript meets the requisite standards for publication in PeerJ. It is my assessment that no further revisions are necessary. I therefore recommend that it be considered for publication in the journal.
Sincerely,

Reviewer 2 ·

Basic reporting

Please find additional comments.

Experimental design

Please find additional comments.

Validity of the findings

Please find additional comments.

Additional comments

ID: peerj-96252-2
Title: Prevalence of anxiety, depression, stress and insomnia among healthcare professionals during COVID-19 in Africa: umbrella review of existing meta-analyses

Thank you for providing a chance to review this manuscript.

Recommendation: Minor revision

Overall, I believe that this study still has some limitations in terms of comprehensiveness and rigor, particularly in the consistency of methodology and the handling of heterogeneity, which might affect the accuracy and generalizability of the results. Additionally, the practical implications of the study’s findings appear to be somewhat delayed and weak. However, as the authors explained in their response letter, this study does indeed provide valuable insights into the mental health issues faced by healthcare professionals in Africa during the COVID-19 pandemic and holds significant pioneering importance. My comments are as follows:

Abstract
Materials and methods: “Stata version 14.0 (StataCorp, College Station, Texas, USA) software was used to analyze the data”, there is no great need to account for this in the summary, and it is suggested that this sentence be deleted.
Conclusions: The language used to present the conclusion is a bit redundant and it is recommended that it be shortened to one sentence.

Introduction
Paragraph 1: This paragraph’s role in the whole introduction is minimal, as the content is repeated in subsequent paragraphs, and the authors need to consider deleting this paragraph or choosing a better expression.
Paragraph 2-5: 1) Multiple references are made to the stress and mental health issues faced by African healthcare workers, and some of the information appears repetitive, weakening the compactness and flow of the argument. Consolidation of repetitive information is recommended and 2 paragraphs are sufficient. 2) Information Overload. A large number of data from different studies are cited, but there is a lack of summarization and synthesis of these data.

Objective
As a demonstration of aim, he is too crude and brief, and suggests appropriate expansion to account for the exact aim of the study.

Materials and materials
Search strategy: 1) Spelling error. “Mendeley removd” should be “Mendeley removed”. 2) “articles published from the start of COVID-19 till March 2023 were included”, indicate in detail the period for which “the start of COVID-19” is to prevail.
Eligibility criteria: Consider cutting the article publication year deadline to 2024, based on the fact that there is already a 16-month gap.

Results
Consider the three-line tabular format.

Discussion
Paragraph 2: The discussion of heterogeneity mentions various factors that may contribute to heterogeneity, such as study populations, methodologies, interventions, etc., but does not give specific examples or argue for the specific impact of these factors in the study. There is a need to analyze more specifically how each factor affects the consistency and generalization of study results.

Thank you and my best,
Your reviewer

Reviewer 3 ·

Basic reporting

No Comments.

Experimental design

There are significant concerns regarding the study design.

1. The author needs to better justify the need for this umbrella review of the existing meta-analyses. While the authors have explained the need for a review of mental health status in Africa, this goal might be better achieved through a meta-analysis of all available studies rather than pooling results from only five existing meta-analyses. This study summarizes five meta-analyses, but the author must justify why these five existing meta-analyses were summarized instead of conducting a new meta-analysis of all existing studies.

2. Additionally, this study includes only five meta-analyses up to March 2023, yet more meta-analyses have likely been published since then. Resource limitations should not compromise the standards expected of a scientific publication.

3. The author should also investigate potential overlaps among the studies included in the five meta-analyses and summarize the percentage of overlapping studies, as these overlaps could bias the conclusions.

Validity of the findings

There are concerns regarding the validity of the findings given the sample size and the potential overlaps of the included meta-analyses. Please see the section 2 for more details.

Reviewer 4 ·

Basic reporting

The revisions have significantly improved the clarity and quality of the manuscript. The additional information provided has strengthened the findings and resolved the issues raised in the initial review. The methodology is sound, the results are robust, and the discussion is comprehensive and well-supported by the evidence presented.

Experimental design

NA

Validity of the findings

NA

---

## Round 0.3 · Minor Revisions

Please can you kindly attend to the comments of the reviewers and send us back another version. If you need any assistance, or any advice or clarification, can you please write to us.

---

## Round 0.4 · accepted · Accept

Well done. I can confirm that:

* You have addressed all of the reviewers' comments and indicated them on the rebuttal section
* I have assessed the revision myself and I am happy with the current version
* Manuscript is ready for publication.